# Synthesis of Porous MgF$_2$ Coating by a Sol–Gel Method Accompanied by Phase Separation

**Yu Lin, Rui Wang \*, Yang Xu, Dongyun Li and Hongliang Ge**

College of Materials and Chemistry, China Jiliang University, Hangzhou 310018, China; linyu19990223@163.com (Y.L.); xuyang22@cjlu.edu.cn (Y.X.)
* Correspondence: wangrui@cjlu.edu.cn

**Abstract:** Surfaces with translucent and wear-resistant effects have a wide range of applications, especially as protective layers. In this work, a simple and convenient method for the preparation of porous magnesium fluoride (MgF$_2$) coatings was proposed. Nano-porous MgF$_2$ powder was prepared with sol–gel and phase separation methods by optimizing the polymer amount and used for the preparation of thick layers onto PVC substrates. The automated surface area and porosity analyzer (BET) and scanning electron microscopy (SEM) confirmed that the layers containing 0.028‰ PEO presented a 3D structure with pore sizes in the range of 16 nm. The layer reached 93% transmittance in the visible region, a Vickers hardness value of 2889.1 kg/mm$^2$, and a friction coefficient of 0.2 in the wear test.

**Keywords:** porous powder; sol–gel method; phase separation; MgF$_2$

## 1. Introduction

Magnesium fluoride (MgF$_2$), a colorless rutile (TiO$_2$) structure and tetragonal crystal system, is widely used as a coating material due to its superior optical properties [1,2]. MgF$_2$ coatings have many excellent properties, including a low refractive index (n = 1.38), a wide transparent band (120 nm~8000 nm), a considerable energy gap (E$_g$ = 11 eV), etc. MgF$_2$ is an ionic compound consisting of many ions with different charges attracted to each other by electrostatic gravity. Therefore, it is very hard and difficult to compress at normal temperature and pressure. In addition, MgF$_2$ [3] has the advantages of high mechanical strength, superior thermal stability, and a high laser damage threshold. It prevents outer material wear damage and prolongs the service life and performance. Therefore, MgF$_2$ coatings have a large number of applications in the preparation of optical coatings. Nowadays, most coating techniques, including vacuum evaporation and magnetron sputtering, are complicated and expensive [4,5]. However, this work presents a simple, low-cost, and environmentally friendly sol–gel wet chemical method for the preparation of MgF$_2$ coatings. The coatings prepared through this method have the advantages of high purity, 3D structure, and high mechanical strength.

The sol–gel process for producing thin coatings dates back to the 20th century [6–8]. Stober [9] et al. synthesized sol through the hydrolysis and condensation of alkyl silicates in the presence of an alkaline catalyst, and silicon dioxide sol particles were created with solvent volatilization and other techniques. Chen [10] et al. used the sol–gel method and PAA template to prepare hollow silica film with 91.3% transmittance at 725 nm. Zhao's group [11] used the sol–gel method to synthesize a TiO$_2$–SiO$_2$ hybrid anti-reflective coating with 90% transmittance at 700 nm. Long [12] et al. used the sol–gel method to prepare high-purity, nanoscale MgF$_2$ films on glass substrates. The transmittance in the central wavelength band (at 700 nm) was 92%. Yang [13] et al. prepared Fe$^{3+}$ doped TiO$_2$ thin films with 92% transmission at 550 nm. Hu [14] et al. used a simple spin-coating method to prepare highly transmissive Ce$^{3+}$ doped TiO$_2$–SiO$_2$ nanocomposite films with 89% transmission at 550 nm. Nakanishi [15] et al. prepared 3D-structured porous silica bulk

with a sol–gel accompanied phase separation method. The organic polymer added in the reaction induced the inorganic sol to undergo spinodal decomposition leading to phase separation. However, the reaction was limited by the precursor activity, thus limiting the application and expansion of the method.

In this study, $MgF_2$ coatings were processed from porous $MgF_2$ powder prepared with sol–gel and phase separation techniques. Magnesium chloride ($MgCl_2$) was used as a precursor to prepare porous $MgF_2$ powders with controlled voids by adding a phase separation inducer during the sol–gel process and controlling the pore structure using phase separation to improve the light transmission of their coatings.

## 2. Materials and Methods

**Materials:** The reagents hydrofluoric acid (HF, AR, 40%), magnesium chloride hexahydrate ($MgCl_2 \cdot 6H_2O$, 99.99% metals basis), propylene oxide (AR, 99.5%), absolute ethanol (AR, 99.9%), ammonia (AR, 28%), and polyethylene glycol (PEO, $M_V$ = 1,000,000) were purchased from Mclean Chemical Reagent Co., Ltd, Los Angeles, CA, USA. These chemicals were used to prepare the porous $MgF_2$ powder. The reagents sodium dodecyl benzene sulfonate, polyvinylidene fluoride (PVDF, $M_V$ = 400,000), and N-Methyl pyrrolidone (NMP, AR, 99.0%) were purchased from Mclean Chemical Reagent Co., Ltd, Los Angeles, CA, USA. They were used to prepare the porous magnesium fluoride coatings.

**Preparation of porous $MgF_2$ powder:** Table 1 lists the raw material components for the preparation of the porous $MgF_2$ powder. Figure 1 shows the preparation process of the porous $MgF_2$ powder. First, a certain amount of PEO was dissolved in a mixture of anhydrous ethanol and deionized water as the solvent. Second, 50 g of $MgCl_2 \cdot 6H_2O$ was dissolved in an appropriate amount of $H_2O$, and then the dissolved PEO solution was mixed with it thoroughly and heated to 80 °C. Third, 30 mL of the mixture of HF and $MgCl_2 \cdot 6H_2O$ was mixed quickly at 80 °C. Then, an amount of catalyst was added, and the temperature was kept at approximately 80 °C for one hour. After the precipitation was completed, the precipitate was stirred again to disperse it, and an amount of inhibitory drying agent was added. Then, the dispersion was aged at 75 °C for at least 3 h. Finally, the sample was dried in an oven at 60 °C to obtain the porous $MgF_2$ product.

**Table 1.** Raw material components for the preparation of porous $MgF_2$ powder.

| Sample | $MgCl_2 \cdot 6H_2O$ (mol%) | HF (mol%) | PEO (mol%) | $H_2O$ (mol%) | EtOH (mol%) | $NH_3 \cdot H_2O$ (mol%) |
|---|---|---|---|---|---|---|
| S−MgF₂:0‰PEO | 2.6 | 1 | / | 6.9 | 2.12 | 0 |
| S | 2.6 | 1 | 0.008‰ | 6.9 | 2.12 | 0 |
| S−MgF₂:0.008‰PEO | 2.6 | 1 | 0.008‰ | 6.9 | 2.12 | 0.02 |
| S−MgF₂:0.02‰PEO | 2.6 | 1 | 0.02‰ | 6.9 | 2.12 | 0.02 |
| S−MgF₂:0.028‰PEO | 2.6 | 1 | 0.028‰ | 6.9 | 2.12 | 0.02 |

**Porous $MgF_2$ characterization:** Crystalline phase information was recognized with X-ray diffraction (XRD, Rigaku Ultima IV, Tokyo, Japan). The instrument was operated at 40 kV and 30 mA current where Cu-K$\alpha$ radiation ($\lambda$ = 1.5406 Å) has been used as an X-ray source with a 2θ range from 10 to 80° scan rate of 5°/min. The surface morphology of S-MgF₂:x‰PEO was characterized with a high-resolution field emission scanning electron microscope (SEM, Hitachi S-4800, Tokyo, Japan) with an accelerating voltage of 5 kV. Fourier transform infrared spectroscopy (FT-IR, Perkin Elmer Spectrum Two, New York, NY, USA) was used to study the functional groups of the obtained S-MgF₂:x‰. The binding energies of Mg, O, and F were investigated with a K$\alpha$ X-ray photoelectron spectroscopy system (XPS, Thermos Scientific K-Alpha, New York, NY, USA). Nitrogen adsorption and desorption curves, pore size distribution, and specific surface area data were obtained with an automated surface area and porosity analyzer (BET, Micromeritics ASAP-2460, New York, NY, USA). The S-MgF₂:x‰ was degassed at 200 °C in a nitrogen atmosphere for 8 h, separately.

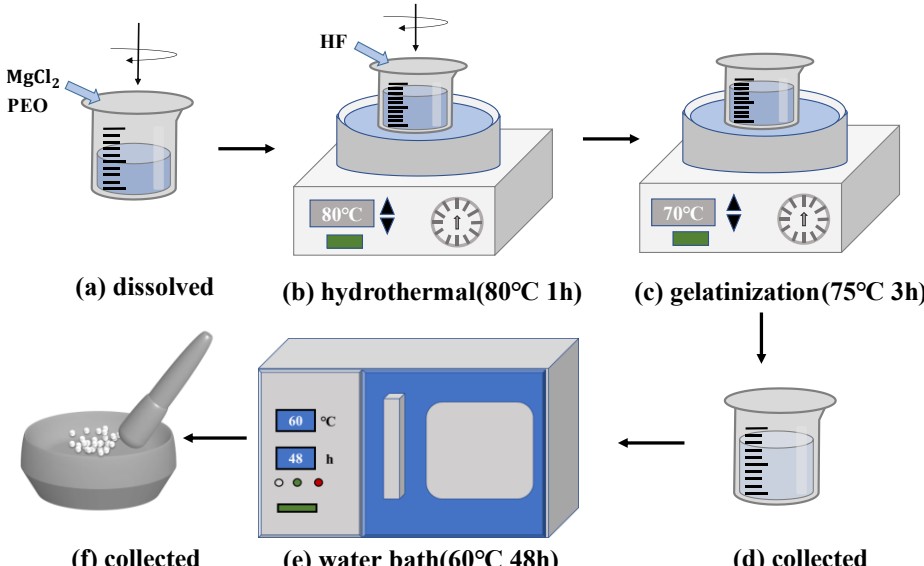

**Figure 1.** Preparation process of porous MgF$_2$ powder.

**Preparation of MgF$_2$ porous coatings:** Table 2 lists the raw material components for the preparation of the porous T-MgF$_2$:x‰ coatings. Figure 2 shows the process for the preparation of the porous T-MgF$_2$:x‰ coatings. At first, the transparent PVC substrates were cleaned until there was no dirt. The PVC substrates were then ultrasonically cleaned with acetone, ethanol, and H$_2$O for 30 min. The cleaned PVC substrates were placed in a drying oven to dry. A quantity of solvent, porous MgF$_2$ powder, surfactant (C$_{18}$H$_{29}$NaO$_3$), and film-forming agent (PVDF and NMP) was added to the beaker to disperse it uniformly in the solution. The dried PVC substrate was placed on the vacuum coating machine. With a pipette gun, an appropriate amount of soluble droplets was absorbed into the center of the PVC substrate, and the rotation speed of the apparatus was adjusted to obtain a coating of proper thickness. The spin-coated PVC substrate was removed, baked on the baking machine at 60 °C, cooled to normal temperature naturally, and then bagged for use.

**Table 2.** Raw material components of porous MgF$_2$ coatings.

| Sample | MgF$_2$ | MgF$_2$ (mol%) | EtOH (mol%) | PVDF (mol%) | NMP (mol%) | C$_{18}$H$_{29}$NaO$_3$S (mol%) |
|---|---|---|---|---|---|---|
| T | / | / | / | 0.05‰ | / | 0 |
| T−MgF$_2$:0.008‰PEO | S−MgF$_2$:0.008‰PEO | 0.3 | 25 | 0.05‰ | 1 | 0.02 |
| T−MgF$_2$:0.02‰PEO | S−MgF$_2$:0.02‰PEO | 0.3 | 25 | 0.05‰ | 1 | 0.02 |
| T−MgF$_2$:0.028‰PEO | S−MgF$_2$:0.028‰PEO | 0.3 | 25 | 0.05‰ | 1 | 0.02 |
| T−MgF$_2$:0‰PEO | / | / | 25 | 0.05‰ | 1 | 0.02 |

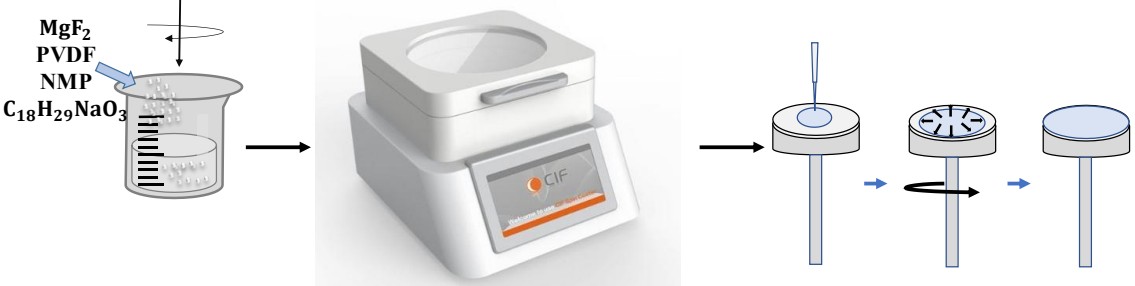

**Figure 2.** Process for the preparation of porous MgF$_2$ coatings.

**MgF$_2$ porous coatings characterization:** The transmittance of the visible band of the samples was observed with a UV spectrophotometer (Shimadzu UV-3600i Plus, Kyoto,

Japan). The hardness of the samples was obtained with a Vickers hardness tester (Innova test Falcon507, Maastricht, The Netherlands). The abrasion resistance was analyzed with a friction and wore tester (Bruker (CETR) UMT-2, Karlsruhe, Germany).

## 3. Results and Discussion

### 3.1. Phase Analysis of Porous MgF$_2$

In this experiment, it was investigated whether PEO could help phase separation. In Figure 3a,b, the two samples S−MgF$_2$:0‰PEO and S are shown. S−MgF$_2$:0‰PEO was not added with PEO, and the microscopic morphology revealed that the particles were very tightly connected. S was added with a small amount of PEO, and the microscopic morphology showed gaps between the particles connected, but it was not obvious. Therefore, it was necessary to consider whether ammonia (NH$_3$·H$_2$O) could be a catalyst. Figure 3b,c shows the microscopic morphology of samples S and S−MgF$_2$:0.008‰PEO. S−MgF$_2$:0.008‰PEO was added with a certain amount of NH$_3$·H$_2$O while sample S was added without it. The gel time of sample S was 2 h; however, S−MgF$_2$:0.008‰PEO had a gel time of 1.5 h. The micrograph reveals that the sample S particles in Figure 3b are tightly connected. Meanwhile, the sample S−MgF$_2$:0.008‰PEO particles of Figure 3c are not tightly connected, and some pore spaces appear between the particles. As illustrated in Equations (1) and (2):

$$MgCl_2·6H_2O + 2HF = MgF_2 + 2HCl + 6H_2O \tag{1}$$

$$NH_3·H_2O + H^+ = NH_4^+ + H_2O \tag{2}$$

NH$_3$·H$_2$O, as an alkaline substance, consumes hydrogen ions faster and raises the pH value rapidly, which reduces the gelation period from 2 h to 1.5 h. At the same time, it promotes phase separation. Then, the influence of the amount of phase separation inducer on the pore structure was investigated. As shown in Figure 3a, the surface microscopic morphology of sample S–MgF$_2$:0‰PEO exhibits very dense particle connections. The color state of the dry gel is shown in Figure 3f, which is translucent white, and this indicates that the particles obtained from this sol–gel preparation are tightly connected. As in Figure 3d, the microscopic morphology of S−MgF$_2$:0.02‰PEO shows some holes between the particles. At this moment, the dry gel color of S–MgF$_2$:0.02‰PEO changed to slightly transparent white, as shown in Figure 3g, which means that phase separation has occurred at this time, and the degree of phase separation has increased. As shown in Figure 3e, S−MgF$_2$:0.028‰PEO presented a 3D pore structure. In addition, the color of the corresponding dry gel is completely white as indicated in Figure 3h, which indicates that the phase separation is complete at this time. Therefore, the phase separation inducer has obvious influence on the microscopic morphology of porous MgF$_2$, and the pore distribution of porous MgF$_2$ can be regulated by the added amount of PEO.

Figure 4a,b shows the nitrogen adsorption–desorption isotherms and Barrett–Joyner–Halenda (BJH) pore size distribution of the porous MgF$_2$ powder for the three samples. All three curves in Figure 4a are similar to the IV-shaped adsorption isotherm proposed by IUPAC, which indicates the presence of multilayer adsorption in all three samples. S−MgF$_2$:0.008‰ has the highest adsorption, which represents the highest number of pores that existed; S−MgF$_2$:0.028‰ has the lowest adsorption, which represents the lowest number of pores that existed. Meanwhile, with the change in the PEO amount, the curve has a large difference, which indicates that the amount of PEO can have a great influence on the phase separation structure. Combined with Figure 4b, many small-diameter mesopores exist in the S−MgF$_2$:0.008‰; the mesopore diameter is approximately 9.3 nm to 10.8 nm, and the specific surface is approximately 889 m$^2$/g. The pore diameter of S−MgF$_2$:0.02% gradually increases to approximately 13.8 nm, and the number of pores decreases with the corresponding specific surface area also decreasing to 123 m$^2$/g. This indicates that the related degree of phase separation is also more intense. The pore size of S−MgF$_2$:0.028‰

rises to approximately 16 nm, and the specific surface area decreases to 65 m$^2$/g, accordingly, as the principle of phase separation is complete. This shows that the amount of PEO can have an impact on the phase separation structure.

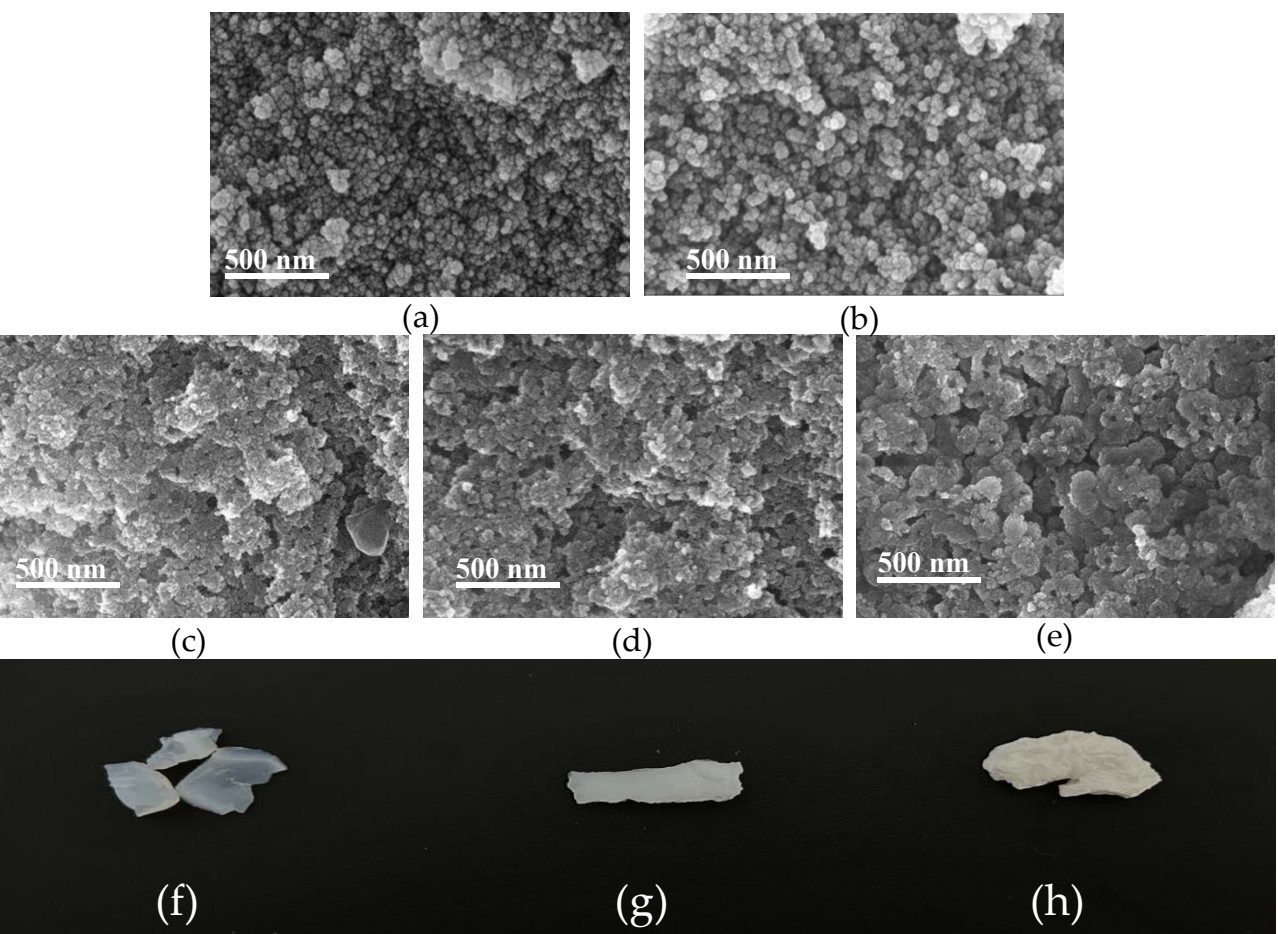

**Figure 3.** (**a**–**e**) Porous MgF$_2$ microscopic morphology. (**a**) S−MgF$_2$:0‰PEO; (**b**) S; (**c**) S−MgF$_2$:0.008‰PEO; (**d**) S−MgF$_2$:0.02‰PEO; (**e**) S−MgF$_2$:0.028‰PEO. (**f**–**h**) Porous MgF$_2$ dry gel. (**f**) S−MgF$_2$:0‰PEO; (**g**) S−MgF$_2$:0.008‰; (**h**) S−MgF$_2$:0.028‰.

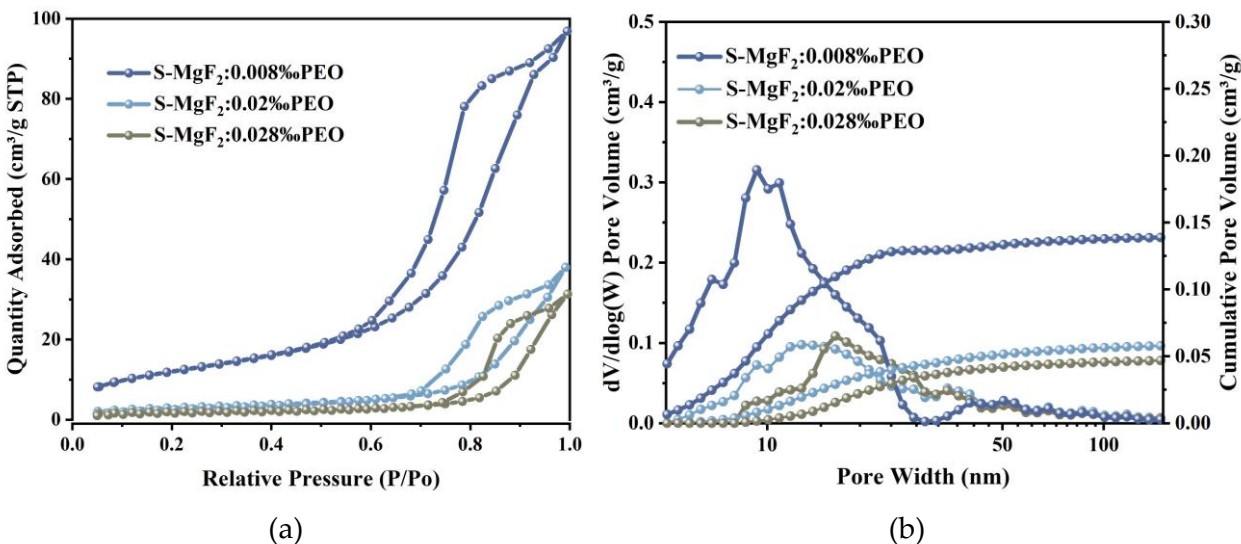

**Figure 4.** (**a**) Nitrogen adsorption–desorption isotherm with different amounts of phase separation inducer; (**b**) BJH pore size distribution of porous MgF$_2$ powder.

### 3.2. Compositional Analysis of Porous MgF$_2$

Figure 5 shows the XRD analysis of the S−MgF$_2$:0‰PEO, S−MgF$_2$:0.008‰PEO, and S−MgF$_2$:0.028‰PEO porous MgF$_2$ powders. The three samples were heat-treated at different temperatures separately to observe whether the PEO addition had an effect on the crystallization. The curves (b,d,f) after 60 °C treatment have characteristic peaks with wide and short peak shapes, which indicates that the crystallization partially occurred. Meanwhile, the characteristic peak of curve (f) is in complete agreement with the standard card of MgF$_2$ (PDF#72-1150), which means the product is pure MgF$_2$. However, curve (b,d) shows the characteristic peaks of ammonium chloride (NH$_4$Cl) when θ = 22.9°, 32.6°, and 46.8°, which is due to the reaction of a small amount of catalyst (NH$_3$·H$_2$O) with Cl$^+$. When the sample is calcined at 500 °C, the impurity NH$_4$Cl decomposes; then, the curve (a,d) characteristic peak is consistent with the MgF$_2$ PDF standard card (PDF#72-1150), which indicates that the product is pure MgF$_2$ without impurities and PEO. The curve (a,c,e) after calcination at 500 °C has a narrow and high peak shape, which means that it has been completely crystallized and is in a stable state at this time. It also reveals that PEO has no effect on the crystallinity of the sample.

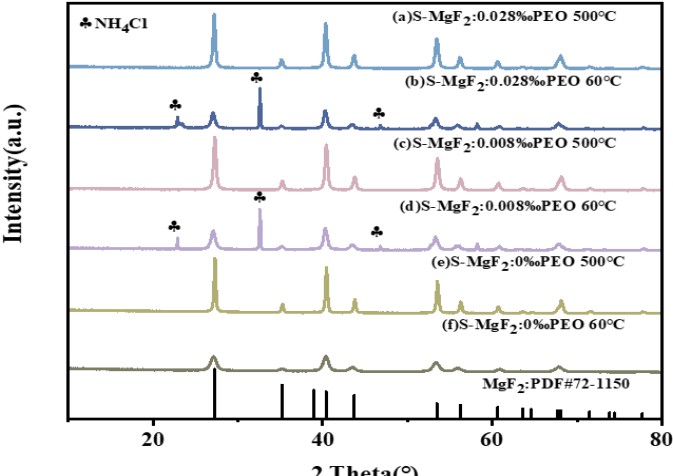

**Figure 5.** XRD analysis of the S−MgF$_2$:0‰PEO, S−MgF$_2$:0.008‰PEO, and S−MgF$_2$:0.028‰PEO powder material at different heat treatment temperatures.

As in Figure 6, the FTIR spectrum of S–MgF$_2$:0‰PEO, S−MgF$_2$:0.008‰PEO, S−MgF$_2$:0.008‰PEO 500 °C, S−MgF$_2$:0.028‰PEO, and S–MgF$_2$:0.028‰PEO 500 °C was in the wavelength range of 500–4000 nm. There are only two distinct IR absorption peaks at approximately 3424 cm$^{-1}$ and 1660 cm$^{-1}$ due to water and ethanol −OH stretching vibrations in the sample S–MgF$_2$:0‰PEO [16,17]. In S–MgF$_2$:0.008‰PEO and S–MgF$_2$:0.028‰PEO, the PEO characteristic peaks appear, which are the C–O bond characteristic peak (1100 cm$^{-1}$), C–C bond stretching vibration peak (847 cm$^{-1}$), and C–H bond stretching vibration peak (2891 cm$^{-1}$), and the formamide characteristic peak appears, which is the C–N bond stretching vibration peak (1403 cm$^{-1}$). After both samples were sintered at 500 °C, all four characteristic peaks disappeared, and there was no PEO in the samples at this time. This also indicates that the process of phase separation is mainly related to the interaction between PEO and the MgF$_2$ oligomers [18]. The MgF$_2$ oligomers are adsorbed on PEO, and the condensation process is induced by PEO. Moreover, with the increase in the PEO amount, the phase separation increases, and the characteristic peaks of the corresponding IR graphs are more obvious. The results show that PEO was successfully doped into MgF$_2$, and PEO was not present in MgF$_2$ after sintering.

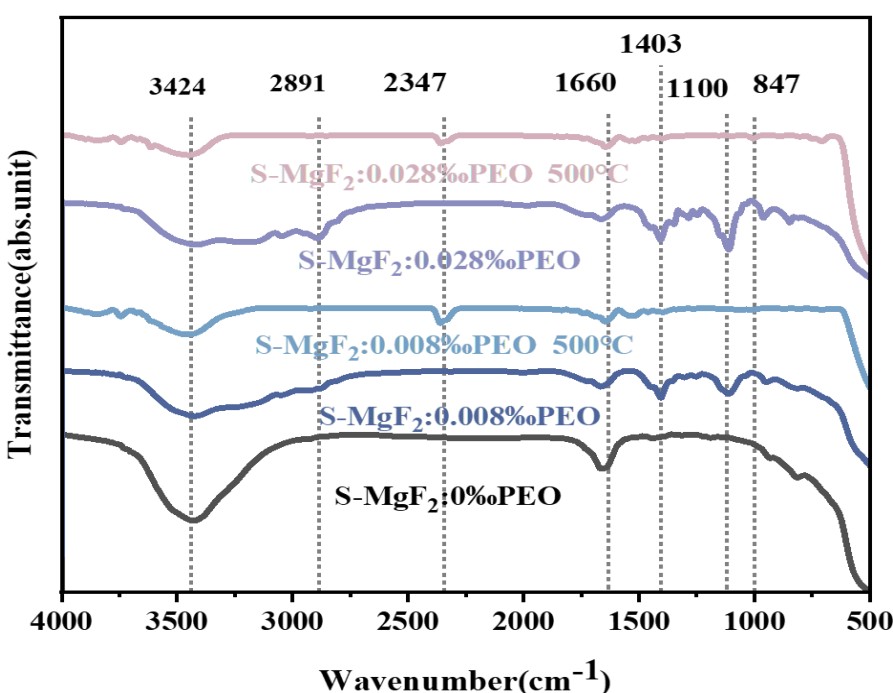

**Figure 6.** Fourier infrared spectrogram of S−MgF$_2$:0‰PEO, S−MgF$_2$:0.008‰PEO, S-MgF$_2$:0.008‰PEO 500 °C, S−MgF$_2$:0.028‰PEO, and S−MgF$_2$:0.028‰PEO 500 °C.

Figure 7a exhibits the XPS full spectrum scans of S−MgF$_2$:0‰PEO, S−MgF$_2$:0.008‰PEO, and S−MgF$_2$:0.028‰PEO. The three peaks of Mg, F, and O can be clearly observed in the figure, and the Mg and F peaks are strong, indicating that the main elements are F and Mg. In Figure 7b–d, the high fractional scans of Mg, F, and O elements are shown. Figure 7b shows the binding energy of 1304.7 eV for Mg of S−MgF$_2$:0‰PEO, which belongs to the Mg atom in the Mg–F–Mg bond [19,20]. It indicates a single peak of S−MgF$_2$:0‰PEO binding energy at 685.2 eV for F, which corresponds to the F1s of F in MgF$_2$ [21]. Meanwhile, the addition of PEO has almost no effect on the Mg and F peaks. Figure 7c shows that the O1s high resolution spectrum contains two peaks associated with the C–O bond (532.38 eV) and O–H bond (533.38 eV). The area of the C–O peak increased gradually with the addition of PEO, which indicates that PEO is successfully involved in the phase separation.

### 3.3. Performance Analysis of Porous MgF$_2$

UV spectrophotometric measures in the visible range were carried out for the sample T,T−MgF$_2$:0.008‰PEO, T−MgF$_2$:0.02‰PEO, and T−MgF$_2$:0.028‰PEO. Figure 8a shows that the T–MgF$_2$:0.028‰PEO coating prepared with the spin-coating method is uniformly dispersed, and the thickness is approximately 5 µm. As shown in Figure 8b, the transmittance of T−MgF$_2$:0.028‰PEO is approximately 93.92% at λ = 725 nm, which is 3% higher than the original substrate sample. T−MgF$_2$:0.028‰PEO has the highest transmittance in several groups of samples and also has the largest addition of PEO in Figure 8b. This shows that the phase separation of porous MgF$_2$ is the highest at this time, and the pore structure formed is the most obvious, which is consistent with the micromorphological response of MgF$_2$ powder [22–24]. The results indicate that the sol–gel method and phase separation technique improved and enhanced the light transmission.

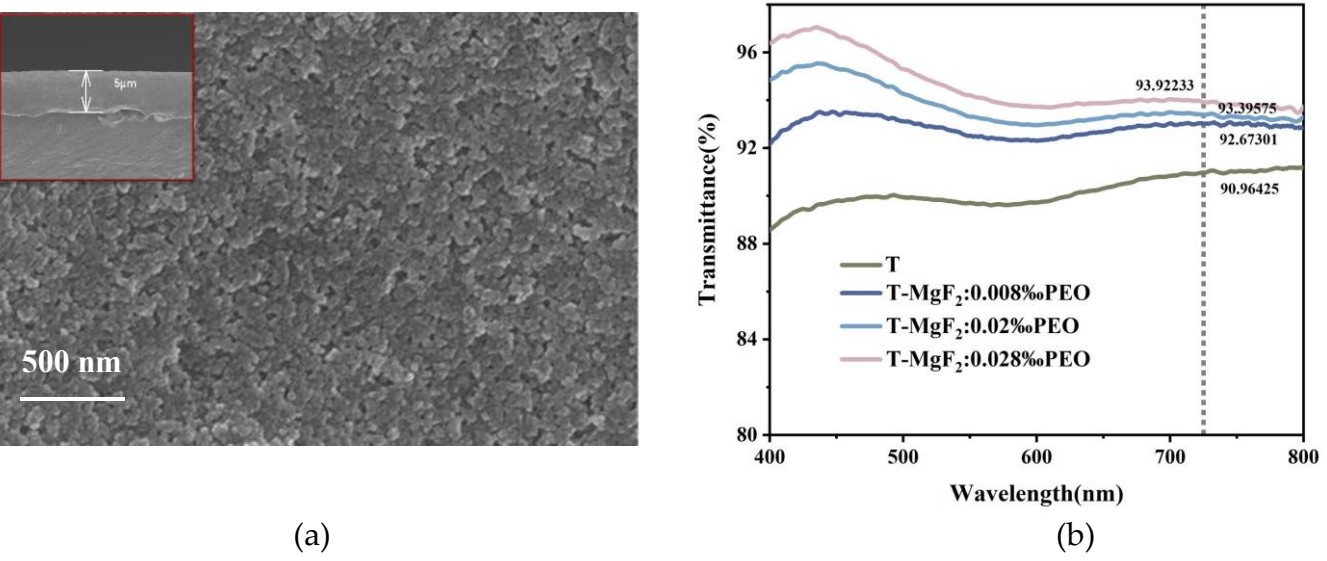

**Figure 7.** XPS analysis of (**a**) S−MgF$_2$:0‰PEO, S−MgF$_2$:0.008‰PEO, and S−MgF$_2$:0.028‰PEO; (**b**) Mg 1 s; (**c**) F 1 s; (**d**) O 1 s.

**Figure 8.** (**a**) Microscopic morphology of sample T−MgF$_2$:0.028‰PEO. (**b**) Transmittance for different ratios of T, T−MgF$_2$:0‰PEO, T−MgF$_2$:0.02‰PEO, and T−MgF$_2$:0.028‰PEO.

For the hardness measurement research, the highest transmittance T−MgF$_2$:0.028‰ PEO was selected. Then, T−MgF$_2$:0.028‰ PEO was used to make T−MgF$_2$:0.028‰ PEO single-layer and T−MgF$_2$:0.028‰ PEO double-layer, substrate T, and a comparison sample T−MgF$_2$:0‰ PEO, which was added to exclude the effect of film-former. The hardness of the coating was studied by taking five test points on each group of coatings by means of the Innova test Europe BV hardness tester, which applied a force of approximately 0.5 N to each test point and held it for 15 s. The indentation is obtained by measuring the diagonal length of the surface indentation. Then, the dimensional hardness value Hv (kg/mm$^2$) is calculated from Equation (3):

$$H_V = \frac{P}{S} = \frac{2P \sin\left(\frac{\theta}{2}\right)}{d} = \frac{18.1855 \times P}{d} \tag{3}$$

P is the load (N), S is the surface area of the indentation (mm$^2$), θ is the opposite angle of the diamond indenter (136°), and d is the average length of the diagonal of the indentation (mm) [25]. The experiment was repeated three times to ensure the accuracy of the results. The Vickers hardness values of the samples were obtained from Equation (1), as shown in Figure 9. The hardness value of the T−MgF$_2$:0.028‰ PEO double-layer coating reached 2995.0 kg/mm$_2$, and the hardness of the T−MgF$_2$:0.028‰ PEO single-layer coating also increased compared with the original substrate sample T. Meanwhile, as a comparison project, the hardness of sample T−MgF$_2$:0‰ PEO without the addition of porous magnesium fluoride was lower than the original substrate and was the lowest. The T−MgF$_2$:0.028‰ PEO single-layer coatings prepared with this method were only approximately 5 μm, and the hardness values obtained were the combined values of the coating and the substrate. In addition, the hardness value of the PVC substrate of 2684.4 kg/mm$_2$ is quite low as well, leading to the measured hardness value of the coating being lower than the actual hardness value. Therefore, the coatings prepared with this method can effectively improve the surface hardness.

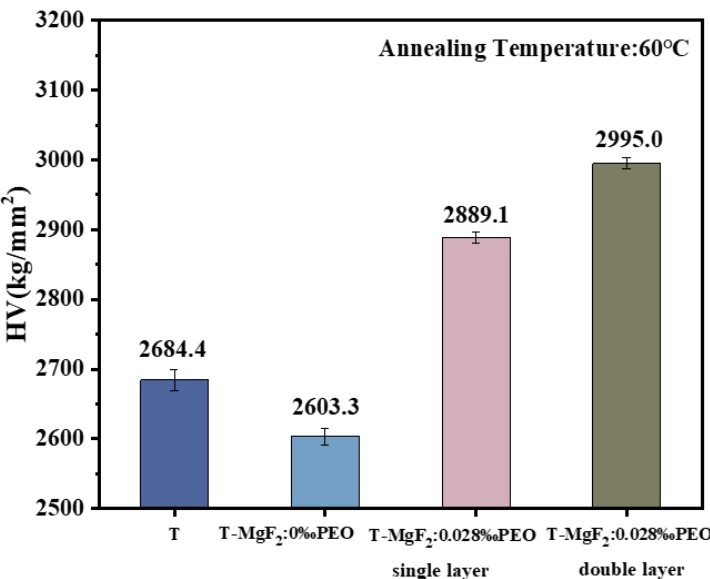

**Figure 9.** Hardness comparison chart of MgF$_2$ coating.

The same four groups of samples as for the hardness test were selected for the experiment. The friction coefficient curves were acquired by applying a linear reciprocating motion of approximately 3 N force to each group of coatings for 30 min by a UMT-2 wear tester from the USA, which is shown in Figure 10a. The masses of the samples both before and after wear were measured with an analytical balance with an accuracy of 10$^{-5}$ g. As depicted in Figure 10b, the wear volume of the sample was acquired with laser confocal

microscopy, a profilometer, and a white light interferometer. The experiment was repeated three times to ensure the accuracy of the results. In Figure 10a, the friction coefficient of the substrate T decreases and then increases during the former 500 s of the friction experiment, and then decreases again after 500 s. The coefficient of friction at this time was as high as 0.4. The friction coefficient curve for the single layer of $T-MgF_2:0.028‰PEO$ single-layer shows a value of approximately 0. 2, which is lower than the coefficient of friction of the original substrate sample T. The result indicates that the friction tester has not yet damaged the coating, and the contact surface never reached the substrate, which shows that the coating played a protective role in the coating formation process. Then, the coefficient of friction obtained for the $T-MgF_2:0.028‰PEO$ double-layer coating is lower than the value of the $T-MgF_2:0.028‰PEO$ single-layer coating, which is approximately 0.1, and further indicates that the coating can reduce the coefficient of friction and achieve the effect of wear resistance. Meanwhile, $T-MgF_2:0‰PEO$ comparison samples were added to exclude the role of coating-forming solutions. The coefficient of friction of its sample $T-MgF_2:0‰PEO$ increased at the beginning, which indicates that the coating was destroyed at this time. The friction coefficient was up to 0.5. Therefore, the protection in the wear test is provided by the addition of porous $MgF_2$ powder. Combining the wear mass shown in Figure 10b, it can be shown that sample $T-MgF_2:0.028‰PEO$ double-layer coating and $T-MgF_2:0.028‰PEO$ single-layer coating have better wear resistance and less loss than the base sample T and the comparison sample $T-MgF_2:0‰PEO$. It is further evidence that the porous $MgF_2$ coating has excellent wear resistance.

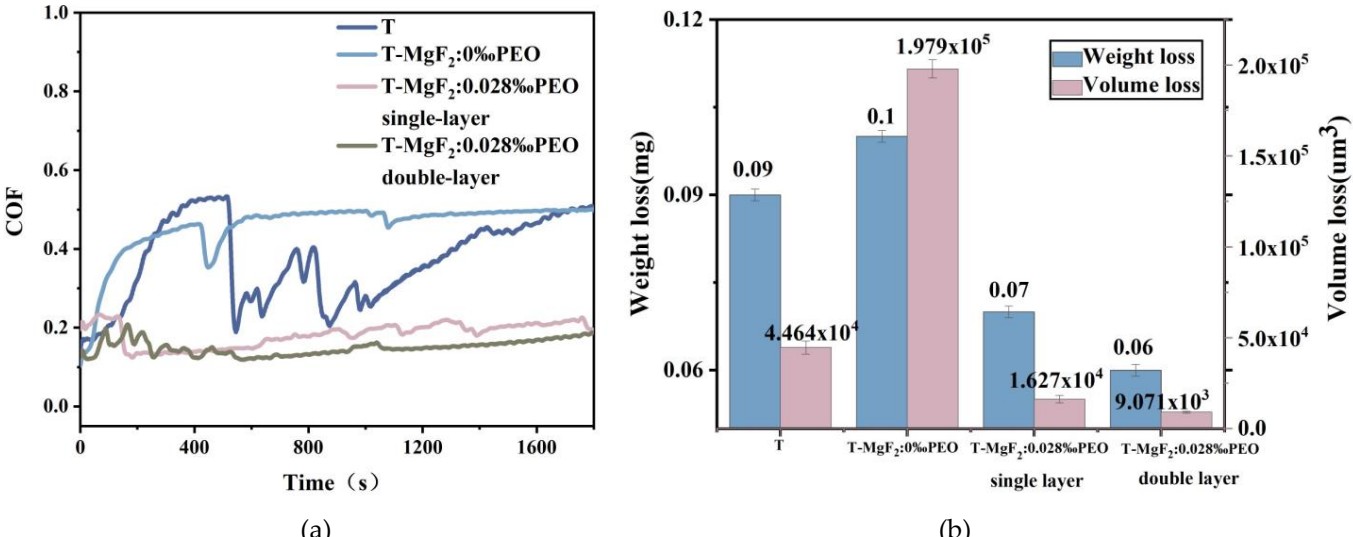

(a)　　　　　　　　　　　　　　　　　　　　　　　(b)

**Figure 10.** (**a**) Comparison of friction coefficients of porous $MgF_2$ coatings; (**b**) Comparative graph of wear mass and volume of porous $MgF_2$ coatings.

## 4. Conclusions

Based on the low refractive index and remarkable optical properties of $MgF_2$, this paper utilized the sol–gel and phase separation methods to prepare nano-porous $MgF_2$ powder with a 3D structure, and the microstructure morphology was significantly changed. In particular, the catalyst ($NH_3 \cdot H_2O$) reduces the gel time and promotes the phase separation, and PEO regulates the $MgF_2$ pore structure. The pore diameter of the champion sample $S-MgF_2:0.028‰PEO$ reached 16 nm. The champion sample $T-MgF_2:0.028‰$ PEO prepared from this powder has a coating thickness of 5 μm and a light transmission of 93% in the visible region, which is 3% higher than the substrate. The $MgF_2$ coatings prepared with the sol–gel and phase separation methods have higher light transmission in the visible region than the nanostructured $MgF_2$ films with the sol–gel method [26–28]. The sample $T-MgF_2:0.028‰$ PEO had a Vickers hardness value of 2889.1 kg/mm$^2$, and the wear resistance test showed a friction coefficient of 0.1, while the mass and volume loss

was minimal. The results indicate that the coating has excellent wear resistance and light transmission properties.

**Author Contributions:** Conceptualization, R.W. and Y.L.; methodology, Y.L.; software, Y.X.; validation, D.L., R.W. and Y.L.; formal analysis, H.G.; investigation, R.W.; resources, Y.X.; data curation, R.W.; writing—original draft preparation, Y.L.; writing—review and editing, R.W.; visualization, D.L.; supervision, H.G.; project administration, R.W. and Y.L.; funding acquisition, R.W. All authors have read and agreed to the published version of the manuscript.

**Funding:** This research received no external funding.

**Institutional Review Board Statement:** Not applicable.

**Informed Consent Statement:** Not applicable.

**Data Availability Statement:** The study did not report any data.

**Conflicts of Interest:** The authors declare no conflict of interest.

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
