# Peer review of "Synthesis of Porous MgF2 Coating by a Sol–Gel Method Accompanied by Phase Separation"

_coatings, doi:10.3390/coatings13061083_

Round 1
Reviewer 1 Report
In this manuscript, the authors present a simple and convenient method for the preparation of porous magnesium fluoride (MgF2) coatings was proposed. The physico-chemical properties of the synthesized samples have been characterized in detail and the hardness and wear characteristics was studied to clarify the results. The results are significant to study the porous coatings, thus I recommend its publication in Coatings. But Prior to its publication, the following issues should be considered.
1. Sol-gel and hydrothermal syntheses are known and widely used methods. The novelty of the work must be clearly addressed and discussed, compare your research with existing research findings and highlight novelty.
2. Authors are advised to add purity of all the precursors used in synthesis process (methods and materials section).
3. Explain what is meant by T and S (tables 3.1 and 3.2)
4. I would recommend that figure 3 be divided into two: on one micrograph of powders, on the other - dry gels.
5. Have X-ray scans been performed on other PEO samples? If so, have characteristic ammonium peaks been observed? I would recommend adding an XRD for the others.
6. I have recommended in Figure 5 to make the distances between diffractograms the same for better clarity.
7. Line 212: "Same time added PEOs have almost no effect on the peaks of Mg and F". But the peak intensity of Mg 1s decreases with increasing % of PEO (Figure 7b). Please explain this.
8. Line 268: “In Figure 10(a), the coefficient of friction of the substrate sample T initially decreased before a little increased, and finally 269 decreased again during the same situation”, but in the figure first there is an increase in the coefficient. Explain what time interval we are talking about.
9. I would recommend that new articles 2022 and 2023 be introduced and discussed.
10. The years of several references are not in bold.

Reviewer 2 Report
The manuscript entitled “A straightforward process for making magnesium fluoride 2 coating with good permeability and resistance to wear” is a well-written and reproducible research manuscript. This manuscript discusses MgF2 coatings that have been processed from porous MgF2 powder prepared by sol-gel and phase separation techniques. In this study, Magnesium chloride (MgCl2) was used as a precursor to prepare porous MgF2 powders and some modifications were made to improve the light transmission of the coatings. However, a few comments needed to be amended to improve the quality of this manuscript.
The comments are below:
1. Title: The title should be presenting the content. Kindly include sol-gel/phase separation technique. Also kindly rephrase the sentence “with good permeability and resistance to wear” in the title.
2. Abstract: wear tests showed an excellent wear resistance.”- please put the result of the wear test rather than putting the unmeasurable adverb.
3. Material & Method: Sentence “All chemical reagents were used without further purification” in the material and method should be removed.
4. Material & Method, Result: Is there any replication of the experiment? If yes, kindly put in the material and method, and also put the error bar in the bar graphs (Figure 9 and 10b).
The language is acceptable. Minor changes needed.
Round 2
Reviewer 1 Report
All comments have been taken into account by the authors.
I recommend the revised manuscript for publication in Coatings